# Assessment of Vulnerability Caused by Earthquake Disasters Based on DEA: A Case Study of County-Level Units in Chinese Mainland

Yuxin Gao [1,2], Xianrui Yu [1,2], Menghao Xi [3] and Qiuhong Zhao [1,2,*]

1 School of Economics and Management, Beihang University, Beijing 100191, China
2 Beijing Key Laboratory of Emergency Support Simulation Technologies for City Operations, Beijing 100191, China
3 School of Emergency Management, Institute of Disaster Prevention, Sanhe 065201, China
* Correspondence: qhzhao@buaa.edu.cn

**Abstract:** Earthquake activity can generate huge energy in a short period of time, bringing enormous risks to people's lives and property safety. This poses a great challenge to regional sustainable development. Meanwhile, due to the complex mechanism, seismic activity is difficult to accurately predict. Therefore, it is of great significance to explore how to reduce earthquake disaster losses from the perspective of human society. In this study, we use vulnerability to reflect the relative impact of earthquake disasters on different counties. The vulnerability caused by earthquakes is calculated with the data envelopment analysis (DEA) method. We use CCR and BCC models to further decompose vulnerability into pure technology vulnerability and scale vulnerability. This study analyzes 69 earthquake disasters that occurred in the Chinese mainland from 2013 to 2020 and explores the influencing factors of pure technology vulnerability from both quantitative and qualitative perspectives. Three main conclusions are drawn. First, four factors, including the added value of the secondary industry, gross domestic product (GDP) per capita, investment density of fixed assets and energy released by earthquakes, have a significant impact on the pure technical vulnerability of counties caused by earthquake disasters. Second, in the samples under consideration, the average vulnerability of the regions with an earthquake magnitude below 5.0 is higher than that of the regions with an earthquake magnitude between 5.0 and 6.0. There are deficiencies in organization, management and facilities in regions with a small earthquake risk. Third, through qualitative analysis, it is shown that the seismic function of buildings affects the vulnerability of counties facing earthquake disasters. The results of the research can provide decision makers with new insights into earthquake prevention and disaster reduction management.

**Keywords:** earthquake disaster; vulnerability decomposition; DEA; county level

## 1. Introduction

Since the beginning of the 21st century, with extreme climate change and geological plate movement, natural and geological disasters have occurred frequently around the world, causing huge losses to the safety of human life and property [1,2]. Earthquake disasters are some of the most harmful geological disasters to the development of human society [3]. It happens very suddenly and is tremendously destructive [4]. This poses great challenges to the sustainable development of the region. On the one hand, earthquake disasters seriously threaten the safety of people's lives. On the other hand, the region suffers huge losses that require a lot of time to rebuild. According to the Emergency Database, from 2000 to 2021, 724,055 people died of earthquakes worldwide, and 121,327,982 people were affected by earthquakes. The total damages caused by earthquakes are as high as USD 569.96 billion [5]. Therefore, how to reduce the adverse effects caused by earthquake disasters is an important issue for human society.

An important research direction is to understand the seismicity itself, such as the mechanism of earthquake occurrence and the characteristics of seismicity [6,7]. Based on this, earthquake prediction, earthquake early warning and other measures can be carried out, which can win more response time for human society. However, seismicity is difficult to accurately predict. The long-term and medium-term prediction is a statistical estimation of the probability of earthquake occurrence, and the short-term prediction needs some precursory phenomena with enough universality and diagnostics [8]. Taking the Tohoku-oki earthquake of the Tohoku region of Japan on 11 March 2011, as an example, although there were many studies on this subduction zone before the earthquake, the knowledge did not make people aware of the potential of an M9.0 earthquake and scenarios considering large near-trench slip in official hazard assessments at Tohoku [9].

In this study, we focus more on the mechanism of earthquake disasters affecting human society and apply the concept of vulnerability to the research of enhancing regional sustainable development capabilities. Vulnerability is a core concept in the fields of climate change, disaster management, public health, development and sustainability science [10]. Adger (2006) [11] pointed out that vulnerability refers to the state of susceptibility to harm from exposure to stresses related to environmental and social changes and a lack of adaptability. Vulnerability reflects the degree to which socio-economic systems in specific regions are more vulnerable to natural disasters [12]. Physical, social, economic and environmental factors may increase the sensitivity of personnel and assets to the hazards. There is an emerging consensus that vulnerability depends to a large extent on the conditions and dynamics of the human environment coupling system exposed to risk. Vulnerability analysis must be comprehensive, dealing not only with the system in question but also with its diverse linkages [13].

The research on vulnerability related to earthquake disasters starts from the vulnerability assessment of physical structures [14–17]. In recent years, factors characterizing regional economic and social conditions have been added to the comprehensive vulnerability assessment, which greatly enriches the vulnerability study [18–24]. Overall, there are empirical methods, mechanical methods and hybrid methods to study vulnerability [25–27]. Our research aims to quantify the vulnerability of the region to earthquake disasters and explore the influencing factors, which belongs to the field of comprehensive evaluation. Due to the complexity of vulnerability, the various quantitative evaluation methods proposed cannot unify the selection of indicators and the setting of weights [28]. Therefore, we choose data envelopment analysis (DEA), a non-parametric method, to calculate vulnerability in a hybrid way.

DEA uses the production frontier to calculate the relative efficiency of decision-making units [29,30]. It has been greatly developed recently and introduced into the field of vulnerability quantitative analysis by scholars [12,31,32]. When a county suffers from an earthquake disaster, it is a disaster-bearing body and a transformation system. Under the influence of earthquake energy, personnel and property losses occur in the county. Vulnerability reflects the ratio of output to input. The principle of the DEA model is similar to the above process, both considering the relative relationship between input and output.

We found that most studies use the annual statistical data of earthquake disasters to calculate the vulnerability of provincial administrative units. It is a type of calculation of the average vulnerability level of a specific region to earthquake disasters. Different from that, we choose the actual earthquake disaster as the decision-making unit (DMU) of the DEA model. Moreover, the statistical data of county-level administrative units are used as the index value. In this way, the calculated vulnerability of counties corresponds to each earthquake disaster, and the subsequent analysis of the influencing factors is more accurate.

This paper has three main purposes. First, we measure the vulnerability of human society in each earthquake disaster, rather than the average vulnerability of a region. Second, the results are further decomposed into pure technology vulnerability and scale vulnerability, which allows us to gain more insights. Third, we analyze the influencing factors of pure technical vulnerability to earthquake disasters from both quantitative and

qualitative perspectives. Pure technical vulnerability is selected as the explanatory variable as it eliminates the impact of county scale, which makes the results of influencing factors widely applicable.

The remainder of this paper is structured as follows. Section 2 introduces the methods and the selection of model variables. In Section 3, the calculation results of vulnerability caused by earthquake disasters are given, and a preliminary descriptive statistical analysis is carried out. Section 4 discusses the influencing factors of pure technical vulnerability in counties from quantitative and qualitative perspectives. Section 5 summarizes the work of the whole article.

## 2. Methodology and Materials

We use DEA method to calculate the vulnerability of the county to earthquake disasters. This is because the DEA method uses the relationship between output and input to establish a model, similar to the process of converting earthquake energy into personnel and economic losses in counties. Moreover, as a non-parametric method, DEA does not require manual setting of indicator weight coefficients, making the calculation process more objective.

### 2.1. Efficiency Decomposition Model Based on DEA

The interaction of environmental and social forces determines exposure and sensitivity. Vulnerability is different in dissimilar locations and systems [33]. In other words, different regions suffering from seismicity with the same parameters may produce separate disaster losses. Therefore, in the process of calculating vulnerability, the influence of earthquake location needs to be considered. DEA method can analyze the vulnerability caused by earthquake disasters through setting them as DMUs and using county-level statistical data, which is closer to the actual situation of the region.

DEA was first proposed by operational research expert Charnes et al. in 1978 [29]. The model is named CCR model, which is taken from the initials of the last names of the three authors. CCR model can estimate the stochastic frontier of effective production using a set of multiple input and multiple output values. Afterward, the efficiency of each DMU can be calculated. Different from the constant return to scale assumption contained in the CCR model, Banker, Charnes and Cooper (1984) developed the BCC model and further discussed the method of calculating efficiency under the assumption of variable return to scale [30]. BCC model is shown as Equation (1), where $\theta_k$ is the efficiency of $DMU_k$, $x$ is the input index, $y$ is the output index, and $\lambda$ is the variable coefficient. It is assumed that there are $m$ input variables, $q$ output variables and $n$ decision-making units in the model.

$$\min \theta_k$$

subject to

$$
\begin{aligned}
\sum_{j=1}^{n} \lambda_j x_{ij} &\leq \theta_k x_{ik} \\
\sum_{j=1}^{n} \lambda_j y_{rj} &\geq y_{rk} \\
\sum_{j=1}^{n} \lambda_j &= 1 \\
\lambda &\geq 0
\end{aligned}
\tag{1}
$$
$$i = 1, 2, \ldots, m;\ r = 1, 2, \ldots, q;\ k = 1, 2, \ldots, n$$

The CCR model assumes that the scale return of production technology remains unchanged, or all evaluated DMUs are in the optimal production scale stage. For the technology of single input and single output, the production frontier under variable return to scale is a curve. All points on the curve are technically effective points. However, the productivity represented by the points is not the same. This is caused by their difference in scales. Therefore, for DMU that is not in the optimal scale production state, the efficiency value calculated by CCR model includes the component of scale inefficiency. BCC model adds $\sum_{j=1}^{n} \lambda_j = 1$ as a constraint that makes the production scale of the projection point

at the same level. The efficiency results of BCC model exclude the influence of scale. The calculation relationship between the three efficiency values is shown in Equation (2).

$$SE = \frac{TE}{PTE} \tag{2}$$

*SE* stands for scale efficiency, which observes the invalid sources of DMUs from the perspective of scale. In the study of vulnerability caused by earthquake disasters, scale efficiency measures the response of county scale to earthquakes. The higher the scale vulnerability is, the more likely population and economic density of the county at that time are to be converted into losses in the earthquake. Technical efficiency (*TE*) is calculated by CCR model and pure technical efficiency (*PTE*) is the result of BCC model.

*2.2. Indexes*

Earthquake disaster is the result of the effect of the huge energy produced by seismic activities on the disaster-bearing system, which usually brings great losses to human society. The variables in the DEA model for measuring vulnerability to earthquake disasters should be selected based on the understanding of this process. In addition, when building the DEA model, we should pay attention to distinguishing the input–output variables of the model and the influencing factors of the vulnerability value. Taking county vulnerability to earthquake disasters as an example, people's knowledge of earthquake prevention and disaster reduction will affect people's response measures when an earthquake occurs. However, it is not a variable in the input and output model. We can regard it as the attribute of disaster-bearing system because these dissimilar conditions make the system show various vulnerabilities.

We investigated some studies using DEA models to measure disaster vulnerability. Huang et al. (2013) [28] constructed input variable indicators from the perspectives of danger of regional hazards and exposure of regional socioeconomic system. The output variables used characterized regional natural disaster loss data. Li et al. (2015) [32] used DEA model to measure the vulnerability of provincial units to geological disasters in China. Input variables include disaster frequency, population density and the sum of GDP. The output variables are the number of casualties and economic losses caused by geological disasters. The study takes provinces as decision-making units and uses the average data from 2004 to 2010 as variable values for calculation. Hou et al. (2016) [34] used super-efficiency DEA model to measure social vulnerability to geological disasters. The model takes the proportion of casualties in the total population and the proportion of economic losses in GDP as output variables. Nine indicators are selected as input variables of DEA model from three perspectives of population, economy and society, such as population density, per capita GDP and medical condition.

From these studies, it can be summed up that the output variables are about earthquake disaster loss, including casualties and economic losses. Input variables can be considered from two perspectives. On the one hand, it should reflect the characteristics or intensity of disasters. On the other hand, it should include the attributes of human society. Based on the above analysis, we use the number of casualties and direct economic losses as the output variables of the DEA model. Energy released by earthquake, population density and GDP density are set as input variables. Variables are shown in Table 1.

**Table 1.** Input–output variables of DEA model for earthquake disaster vulnerability.

| Variables | Code | Definition and Measurement |
|---|---|---|
| Energy | $x_1$ | Energy released by seismicity. It is obtained by the Gutenberg–Richter empirical function |
| Population density | $x_2$ | Average population per square kilometer |
| GDP density | $x_3$ | Average gross domestic product per square kilometer |
| Number of casualties | $y_1$ | Number of injured and dead people |
| Direct economic loss | $y_2$ | Material damage caused by earthquakes and secondary disasters |

Magnitude is a commonly used parameter to describe seismic characteristics. However, magnitude does not directly indicate the energy released by seismicity. We use the Gutenberg–Richter empirical function to convert the magnitude into the energy released by the earthquake, so as to meet the DEA method's requirement that different DMUs can be added linearly on each variable. The Gutenberg–Richter empirical function is shown in Equation (3).

$$lgE = 1.5 \times M_s + 11.8 \tag{3}$$

where $E$ refers to the energy released by seismicity, and $M_s$ is surface wave magnitude, which is recorded by humans [35].

When calculating the vulnerability of counties suffering from earthquake, the indexes shown in Table 1 are used in Equation (1). Given the values of input variable $x_{ij}(i = 1, 2, \ldots, m)$ and output variable $y_{rj}(r = 1, 2, \ldots, q)$, we can obtain $\lambda_j(j = 1, 2, \ldots, n)$ and $\theta_k$. $\theta_k$ represents vulnerability.

### 2.3. Data Processing

This study investigates the earthquake disasters that occurred in Chinese mainland from 2013 to 2020. According to the annual earthquake activity data released by the official website of the China Earthquake Administration, the Chinese mainland had 99 earthquake disasters (including the 2015 Nepal earthquake which influenced Tibet), involving 17 provinces, autonomous regions and municipalities directly under the Central Government. Among them, 23 earthquakes occurred in Sichuan Province, 21 in Yunnan Province and 19 in Xinjiang Province. The three provinces accounted for 63.6% of the total number of earthquake disasters.

Under the background of earthquake disasters, we select the county-level administrative unit corresponding to the disaster as disaster-bearing system. The announced epicenter location of earthquake disasters can generally be accurate to county-level administrative units. In terms of its main scope of influence, it is concentrated in county-level administrative units. As we want to measure the vulnerability caused by earthquakes, rather than the vulnerability of the region itself, we can obtain more accurate vulnerability results of earthquake disasters by selecting county-level administrative units as the disaster-bearing system.

We screen the samples according to 2 considerations. The first is data integrity. The empirical analysis part needs statistical data to support it, so samples lacking earthquake loss data and samples unable to query county population and economic statistical information are deleted. The second consideration is that the data are supposed to represent the situation of counties when they suffer earthquake disasters. In this study, the statistical data of the county in the previous year is used as the indicator value. Therefore, for counties with two or more earthquakes in the same year, only the first earthquake disaster in the same year is retained in the sample. In addition, since the epicenter of the Nepal earthquake is not within the scope of Chinese mainland, and the impact of the earthquake on China is counted by provinces, the earthquake is also removed from the sample. Finally, 69 valid samples are obtained in this study. The sample contains 16 provinces, including Anhui,

Gansu, Guangxi, etc. Among them, 43 earthquakes occurred in Sichuan, Xinjiang and Yunnan, accounting for 62.3% of the total samples.

## 3. Results and Analysis

In this paper, the DEA method described in Section 2.1 is used to calculate the vulnerability of the county to earthquake disasters. The input variables of the model are energy, population density and GDP density. The output variables of the model are the number of casualties and direct economic loss. MaxDEA 8.0 Ultra software is used to solve the DEA model and obtain the vulnerability values. The earthquake data are released by the official website of the China Earthquake Administration. The statistical data on the population and economy in counties are from the China County Statistical Yearbook from 2013 to 2020. Since the energy released by seismicity is an objective attribute of an earthquake, this variable is set as a variable that cannot be controlled at will [36]. In addition, the change in vulnerability can be promoted from both the other input variables and output variables. Hence, we adopt the unguided model setting. The calculation results are shown in Table 2.

Vulnerability value is the relative efficiency of each disaster-bearing system to convert seismic energy, population density and economic density into casualties and direct economic losses. From the dual programming of the CCR model or the BCC model, it can be determined that the objective function is to obtain the maximum efficiency of the DMU. In the context of vulnerability, we calculate the most unfavorable vulnerability value for the disaster-bearing system. Table 2 decomposes the sources of vulnerability into scale vulnerability and pure technology vulnerability. The scale vulnerability values present the difference in vulnerability caused by the scales of counties under the assumption of variable scale return. The pure technology vulnerability values reflect the vulnerability difference caused by technology, management, organization and other factors.

Earthquake magnitude is an important parameter to reflect the energy released by seismic activity. We divided the samples into three groups according to the magnitude level and expected to get more detailed findings. In the sample, there are 19 earthquake disasters of M ≤ 5.0, 34 earthquake disasters of 5.0 < M ≤ 6.0, and 16 earthquake disasters of M > 6.0. Earthquake disasters with 5.0 < M ≤ 6.0 account for 49.3% of the total samples. Table 3 shows the average value of vulnerability and input–output variables corresponding to the earthquake disasters at the magnitude level of each group.

It can be found from Table 3 that earthquake disaster groups with different magnitude levels show different characteristics of pure technical vulnerability and scale vulnerability. From the perspective of average value, the group of earthquakes with 5.0 < M ≤ 6.0 has the lowest level of pure technology vulnerability and scale vulnerability. This shows that counties suffering from earthquakes with 5.0 < M ≤ 6.0 can generally better cope with earthquake disasters. From the perspective of input variables, the scale of these counties is at a medium level. In the other two groups, the counties suffering from earthquake disasters with M > 6.0 have the highest level of scale vulnerability, while those suffering from earthquake disasters with M ≤ 5.0 have the highest level of pure technical vulnerability. The scale vulnerability is mainly related to the comparison between the input quantities of DMUs. As the average population and economic density of counties suffering from earthquakes with M > 6.0 are the lowest, their scale vulnerability is mainly affected by the energy released by the earthquake. Although the counties suffering from earthquakes with M ≤ 5.0 has the lowest loss, their overall pure technical vulnerability level is the highest. The results show that compared with other counties, the loss conversion level of counties suffering from earthquakes below M5.0 is higher. If the counties do not improve their ability of earthquake prevention and disaster reduction as soon as possible, unnecessary human and economic losses will be caused.

**Table 2.** Calculation results of earthquake disaster vulnerability.

| Date | Epicenter Location | PTV | SV | Date | Epicenter Location | PTV | SV | Date | Epicenter Location | PTV | SV |
|---|---|---|---|---|---|---|---|---|---|---|---|
| 26 June 2020 | Yutian, Xinjiang | 0.128 | 0.161 | 11 May 2017 | Kuershi, Xinjiang | 1.000 | 1.000 | 14 March 2015 | Yingquan, Anhui | 1.000 | 1.000 |
| 18 May 2020 | Qiaojia, Yunnan | 1.000 | 1.000 | 27 March 2017 | Yangbi, Yunnan | 0.336 | 0.011 | 1 March 2015 | Cangyuan, Yunnan | 0.265 | 0.924 |
| 1 April 2020 | Shiqu, Sichuan | 0.253 | 0.194 | 27 December 2016 | Rongchang, Chongqing | 0.085 | 0.001 | 22 February 2015 | Shawan, Xinjiang | 0.956 | 0.003 |
| 19 January 2020 | Jiashi, Xinjiang | 0.026 | 0.987 | 20 December 2016 | Qiemo, Xinjiang | 1.000 | 0.196 | 14 January 2015 | Jinkouhe, Sichuan | 1.000 | 0.908 |
| 16 January 2020 | Kuche, Xinjiang | 0.008 | 0.053 | 14 December 2016 | Ruoqiang, Xinjiang | 1.000 | 0.039 | 10 January 2015 | Atushi, Xinjiang | 0.307 | 0.003 |
| 26 December 2019 | Yingcheng, Hubei | 0.058 | 0.001 | 8 December 2016 | Hutubi, Xinjiang | 0.136 | 0.731 | 7 October 2014 | Jinggu, Yunnan | 0.164 | 0.796 |
| 18 December 2019 | Zizhong, Sichuan | 0.052 | 0.026 | 25 November 2016 | Aketao, Xinjiang | 0.128 | 0.737 | 25 October 2014 | Wencheng, Zhejiang | 1.000 | 1.000 |
| 25 November 2019 | Jingxi, Guangxi | 0.005 | 0.055 | 17 October 2016 | Zaduo, Qinghai | 0.478 | 0.420 | 1 October 2014 | Yuexi, Sichuan | 0.579 | 0.004 |
| 28 October 2019 | Xiahe, Gansu | 0.046 | 0.089 | 23 September 2016 | Litang, Sichuan | 0.335 | 0.022 | 3 August 2014 | Ludian, Yunnan | 1.000 | 1.000 |
| 16 September 2019 | Ganzhou, Gansu | 0.004 | 0.007 | 11 August 2016 | Dianjiang, Chongqing | 0.361 | 0.397 | 5 April 2014 | Yongshan, Yunnan | 0.505 | 0.806 |
| 8 September 2019 | Weiyuan, Sichuan | 0.421 | 0.870 | 31 July 2016 | Cangwu, Guangxi | 0.013 | 0.099 | 12 February 2014 | Yutian, Xinjiang | 1.000 | 1.000 |
| 3 January 2019 | Gong, Sichuan | 0.001 | 0.084 | 18 May 2016 | Yunlong, Yunnan | 1.000 | 0.003 | 16 December 2013 | Badong, Hubei | 0.044 | 0.020 |
| 16 December 2018 | Xingwen, Sichuan | 0.002 | 0.452 | 11 May 2016 | Dingqing, Tibet | 1.000 | 0.964 | 1 December 2013 | Keping, Xinjiang | 0.256 | 0.027 |
| 31 October 2018 | Xichang, Sichuan | 0.001 | 0.099 | 12 March 2016 | Yanhu, Shanxi | 1.000 | 0.960 | 23 November 2013 | Qianguo, Jilin | 0.320 | 0.060 |
| 11 October 2018 | Zigui, Hubei | 0.840 | 0.000 | 11 February 2016 | Xinyuan, Xinjiang | 0.293 | 0.003 | 31 August 2013 | Shangri-la, Yunan | 0.216 | 0.508 |
| 8 September 2018 | Mojiang, Yunnan | 0.024 | 0.964 | 21 January 2016 | Menyuan, Qinghai | 0.049 | 0.989 | 12 August 2013 | Zuogong, Tibet | 1.000 | 1.000 |
| 4 September 2018 | Jiashi, Xinjiang | 0.022 | 0.460 | 14 January 2016 | Luntai, Xinjiang | 0.050 | 0.084 | 22 July 2013 | Min, Gansu | 1.000 | 0.906 |
| 6 May 2018 | Chengduo, Qinghai | 0.390 | 0.099 | 30 October 2015 | Changning, Yunnan | 0.454 | 0.006 | 22 April 2013 | Keerqin, Inner Mongolia | 0.017 | 0.872 |
| 18 November 2017 | Milin, Tibet | 0.611 | 0.894 | 3 July 2015 | Pishan, Xinjiang | 1.000 | 1.000 | 20 April 2013 | Lushan, Sichuan | 1.000 | 1.000 |
| 30 September 2017 | Qingchuan, Sichuan | 0.016 | 0.033 | 22 May 2015 | Rushan, Shandong | 0.354 | 0.775 | 29 March 2013 | Changji, Xinjiang | 0.004 | 0.078 |
| 9 August 2017 | Jinghe, Xinjiang | 0.416 | 0.710 | 15 April 2015 | Alashan, Inner Mongolia | 0.202 | 0.802 | 11 March 2013 | Atushi, Xinjiang | 0.105 | 0.018 |
| 8 August 2017 | Jiuzhaigou, Sichuan | 0.758 | 0.797 | 15 April 2015 | Lintao, Gansu | 1.000 | 1.000 | 3 March 2013 | Eryuan, Yunnan | 0.021 | 0.740 |
| 16 June 2017 | Zigui, Hubei | 1.000 | 0.000 | 30 March 2015 | Jianhe, Guizhou | 0.014 | 0.607 | 18 January 2013 | Baiyu, Sichuan | 0.205 | 0.117 |

**Table 3.** Average value of vulnerability and input–output variables grouped by magnitude level.

| Index and Variable | M ≤ 5.0 | 5.0 < M ≤ 6.0 | M > 6.0 |
|---|---|---|---|
| Pure technical vulnerability | 0.676 | 0.224 | 0.556 |
| Scale vulnerability | 0.374 | 0.336 | 0.821 |
| Energy | 11.876 | 129.704 | 9401.304 |
| Population density | 257.186 | 102.355 | 44.359 |
| GDP density | 8997.495 | 3430.398 | 679.127 |
| Number of casualties | 5.368 | 14.294 | 1306.750 |
| Direct economic loss | 5800.789 | 42,805.329 | 899,471.938 |

## 4. Discussion

The vulnerability of counties is varied, caused by different earthquake disasters, which is related to the characteristics of earthquake disasters and counties. Therefore, it is necessary to discuss the main influencing factors of vulnerability. In addition, we noticed that the average level of pure technical vulnerability caused by earthquake disasters with M ≤ 5.0 is higher than that of the group with M > 6.0. This phenomenon is also discussed from the perspective of the seismic performance of buildings.

### 4.1. Influencing Factors of Vulnerability Caused by Earthquakes

There are few studies that directly study the influencing factors of earthquake disaster vulnerability. When selecting the influencing factors, we refer to the relevant research on the vulnerability index system. De Ruiter et al. (2017) [37] classified vulnerability indicators as physical indicators and social indicators. Physical indicators are directly related to the characteristics of the exposed assets. Social indicators include demography, consciousness, social economics and institutional factors. Li et al. (2015) [32] calculated the quantitative value of geological disaster vulnerability in various provinces of China. The authors selected six variables from three perspectives: natural factors, economic development factors and human control factors. Considering the availability of data, in the quantitative analysis part, this study focuses on exploring the impact of socio-economic factors on the vulnerability of counties to earthquake disasters. Considering that the vulnerability of counties caused by different levels of earthquake disasters is obviously different, this section will also discuss whether there is a significant relationship between the energy released by earthquakes and the vulnerability caused by them. Eight factors are finally selected for the analysis, as shown in Table 4.

**Table 4.** Definition and measurement of potential influencing factors.

| Variables | Definition and Measurement |
|---|---|
| Added value of primary industry | The results of the production activities of the primary industry conducted by all permanent resident units in the region within one year at market prices. |
| Added value of secondary industry | The results of the production activities of the secondary industry conducted by all permanent resident units in the region within one year at market prices. |
| Added value of tertiary industry | The results of the production activities of the tertiary industry conducted by all permanent resident units in the region within one year at market prices. |
| Per capita GDP | Per capita GDP in counties. |
| Investment density in fixed assets | Fixed asset investment per square kilometer. |
| Urbanization rate | The proportion of local non-agricultural population in the permanent population. |
| Junior high school enrollment rate | The proportion of junior middle school graduates who continue to study in ordinary high schools or secondary vocational schools. |
| Energy released by earthquake | The energy corresponds to the magnitude of the earthquake, which is obtained by the Gutenberg–Richter empirical function. |

Relevant statistical data were obtained from the official websites of the China Earthquake Administration and the National Bureau of Statistics and the China county statistical

yearbook. Before entering the regression model, we standardized the data. The formula is shown in Equation (4).

$$x_s = \frac{x - x_{min}}{x_{max} - x_{min}} \tag{4}$$

where $x$ is the original data of the variable, $x_s$ is the standardized data, $x_{min}$ is the minimum value of the variable in all samples, and $x_{max}$ is the maximum value of the variable in all samples.

Since the vulnerability value is in the range of 0–1, we use a Tobit regression model to test the relationship between potential influencing factors and vulnerability. Here, the explained variable uses pure technical vulnerability to reduce the impact of county-scale heterogeneity. In this way, the county influencing factors passing the test are not affected by the regional scale, which has more universal practical significance. Based on the analysis of vulnerability-influencing factors, we establish the regression equation according to Equation (5).

$$
\begin{aligned}
\textit{pure technical vulnerability} &= \alpha + \beta_1 \times \textit{value primary} + \beta_2 \times \\
\textit{value secondary} &+ \beta_3 \times \textit{value tertiary} + \beta_4 \times \textit{per capital GDP} + \\
\beta_5 &\times \textit{investment density in fixed assets} + \beta_6 \times \textit{urbanization rate} + \\
&\beta_7 \times \textit{enrollment rate} + \beta_8 \times \textit{energy} + \varepsilon
\end{aligned}
\tag{5}
$$

We use Stata 17.0 to test the equation, and the results are shown in Table 5.

**Table 5.** Model test results of regression equation.

| Variable | Coefficient | Std. Err. | t | $p > |t|$ |
|---|---|---|---|---|
| Added value of primary industry | −0.019 | 0.365 | −0.05 | 0.958 |
| Added value of secondary industry | −1.578 ** | 0.620 | −2.55 | 0.013 |
| Added value of tertiary industry | 0.219 | 0.550 | 0.40 | 0.691 |
| Per capita GDP | 1.412 ** | 0.648 | 2.18 | 0.033 |
| Investment density in fixed assets | 0.854 * | 0.440 | 1.94 | 0.057 |
| Urbanization rate | −0.229 | 0.276 | −0.83 | 0.410 |
| Junior high school enrollment rate | −0.241 | 0.256 | −0.94 | 0.349 |
| Energy released by earthquake | 1.207 * | 0.652 | 1.850 | 0.069 |
| Constant | 0.617 *** | 0.191 | 3.22 | 0.002 |

\* $p < 0.10$, ** $p < 0.05$, *** $p < 0.01$. (Two-tailed test).

From Table 5, we can derive that the added value of the secondary industry variable is significant at the level of 0.05, and the coefficient is negative (−1.578, $p = 0.013$). This shows that increasing the added value of the secondary industry in the county is conducive to reducing the vulnerability of counties to earthquake disasters. The variable of per capita GDP has a positive impact (1.412, $p = 0.033$) on the vulnerability of counties caused by earthquake disasters. In other words, under other conditions unchanged, the higher the per capita GDP, the greater the pure technical vulnerability to earthquake disasters in counties. The impact of investment density in fixed assets on vulnerability to earthquake disasters in counties is statistically significant, and the impact is also positive (0.854, $p = 0.057$). These three variables are related to economic development. Another variable that has a significant impact on vulnerability is the energy released by earthquakes (1.207, $p = 0.069$). It gives evidence of the earthquake characteristics.

The added value of the secondary industry can reflect the structure of economic development. The secondary industry includes mining, manufacturing, power, heat, gas and water production and supply, and construction. Li et al. (2015) [32] found that the industrial growth rate is the most significant factor affecting vulnerability to regional geological disasters, and industrial development can reduce vulnerability to geological disasters. This is consistent with the conclusion that an increase in the added value of the secondary industry can reduce the vulnerability of a county to earthquake disasters. The main reason is that the development of the secondary industry can promote the

upgrading of infrastructure, housing and other buildings. On the one hand, the enterprises related to the secondary industry tend to have a large scale and are subject to more strict seismic fortification supervision. On the other hand, the continuous development of the construction industry can also contribute to the practical application of seismic technology in construction engineering.

Per capita GDP is an important indicator of regional economic development and people's living standards. Huang et al. (2013) [28] found that there is a significant negative correlation between per capita GDP and regional natural disaster vulnerability. That is, an increase in per capita GDP will reduce vulnerability to regional natural disasters. However, Li et al. (2015) [32] concluded that the per capita GDP is positively related to vulnerability to regional geological disasters. The results of this study also show that, under other conditions unchanged, the growth of per capita GDP will enhance the vulnerability caused by earthquake disasters in counties. For different kinds of vulnerability, the growth of per capita GDP has brought about the impact of strengthening or weakening. This shows that the relationship between per capita GDP and vulnerability is affected by disaster types. In addition, Toya and Skidmore (2007) [38] pointed out that an increase in income enhanced people's demand for security. Higher incomes enable people to invest in more expensive preventive measures to deal with natural disasters. In this study, with the growth of per capita GDP, the vulnerability caused by earthquake disasters becomes higher. This proves that there is an inadaptability between the level of earthquake disaster prevention and the economic development in counties.

The energy released by the earthquake represents the strength of seismic activity. The stronger the earthquake activity, the more destructive energy the disaster-bearing body will suffer. According to the results in Table 5, when suffering from more intense seismic activity, the county often shows higher vulnerability. Xinjiang Uygur Autonomous Region is an earthquake-prone area. In 2018 and 2020, earthquakes of a magnitude of 5.5 and 6.4 occurred in Jiashi County. From the statistical data, the added value of the secondary industry, per capita GDP and investment density in fixed assets of Jiashi County corresponding to the two earthquakes are relatively close. Their pure technical vulnerabilities are 0.022 and 0.026. This reflects that vulnerability caused by high-magnitude earthquakes is stronger. To some extent, the characteristics of seismic activity affect the vulnerability of human society.

### 4.2. Effect of Seismic Function of Buildings on Vulnerability Caused by Earthquakes

The social system of the county has an important impact on its vulnerability to earthquakes. In addition, the seismic capacity of infrastructure in counties also plays an important role in vulnerability. The improvement of the seismic performance of buildings can minimize the physical damage caused by earthquakes [39]. However, the actual seismic performance of buildings is affected by many factors, such as technology, structure, materials and building years. It is also difficult to obtain accurate quantitative description data of buildings in counties. Therefore, we use specific earthquake disaster cases to qualitatively analyze the relationship between the seismic performance of buildings and the earthquake disaster vulnerability in counties.

The analysis of Table 3 makes us note that the earthquake disaster group with the lowest magnitude of M ≤ 5.0 has the largest average pure technical vulnerability. Among the 19 earthquake disasters, the pure technical vulnerability caused by 9 earthquakes reached 1.000, accounting for nearly 50%. The earthquakes are generally distributed in low-risk areas such as Zhejiang Province, Anhui Province and Hubei Province. This shows that compared with other regions, under the same input level, the output of these regions is on the higher side. Generally speaking, the probability of serious earthquake disasters in areas with low earthquake risk is relatively lower, and from the perspective of the impact factors of pure technical vulnerability, the lower the magnitude of the earthquake, the smaller the vulnerability of the county should be. However, the anomalies in the results further remind us of the importance of continuously enhancing the ability of human society

to cope with earthquake disasters. We should not relax our vigilance in areas with low earthquake risk. We selected three representative earthquake disasters for analysis, and the relevant data are shown in Table 6.

**Table 6.** Representative cases of earthquake disasters with M $\leq$ 5.0.

| Date | Epicenter | Magnitude | $x_1$ | $x_2$ | $x_3$ |
|---|---|---|---|---|---|
| 18 May 2020 | Qiaojia, Yunnan | 5 | 19.953 | 172.563 | 2830.313 |
| 14 March 2015 | Yingquan, Anhui | 4.3 | 1.778 | 911.076 | 17,736.225 |
| 25 October 2014 | Wenchen, Zhejiang | 4.2 | 1.259 | 306.110 | 4534.416 |

In the three earthquake disasters mentioned in Table 6, most houses in Qiaojia County are built on hillsides or river valleys due to topographic factors, which makes them more vulnerable to geological secondary disasters caused by earthquakes. Zhang et al. (2020) [40] found that the earthquake damage index of houses in Qiaojia County is higher than that of other regions in Yunnan. The Yingquan M4.3 earthquake was more destructive than any previous earthquake of the same magnitude in the region. One important reason is that the focal point was relatively shallow, and the other direct reason is the damage to buildings caused by the earthquake [41]. The earthquake caused the "Roman columns" (nonstructural building components) used for decoration on the second floor of the rural residence to fall to varying degrees, which had an adverse impact on the safety of the residents' lives and property [42]. On 25 October 2014, Wencheng County experienced an M4.2 earthquake, which was also the largest earthquake in the series of earthquake swarm activities since 12 September. The epicenter of the earthquake was in a mountainous area where the seismic fortification was weak. The houses in the disaster area cannot withstand continuous earthquakes and suffer relatively serious losses. The above analysis shows that the poor seismic performance of buildings is an important reason for high vulnerability in the case of earthquakes with low magnitude.

Among the earthquake disasters with higher magnitude, we noticed that the magnitude and maximum intensity of the Lushan earthquake in 2013 and the Jiuzhaigou earthquake in 2017 of Sichuan Province were the same, but there were significant differences in vulnerability, as shown in Table 7.

**Table 7.** Representative cases of earthquake disasters with M $\leq$ 5.0.

| Date | Epicenter | Magnitude | $x_1$ | $x_2$ | $x_3$ | $y_1$ | $y_2$ | PTV | SV |
|---|---|---|---|---|---|---|---|---|---|
| 20 April 2013 | Lushan, Sichuan | 7.0 | 19,952.623 | 88.682 | 1857.786 | 13,215 | 6,651,370 | 1.000 | 1.000 |
| 8 August 2017 | Jiuzhaigou, Sichuan | 7.0 | 19,952.623 | 15.361 | 493.740 | 573 | 804,300 | 0.758 | 0.797 |

In these two earthquake disasters, the occurrence time of the Lushan earthquake and the Jiuzhaigou earthquake is 8:02 and 21:19, respectively. When the earthquake occurred, the local people were generally living in houses. From this, we can know that the seismic performance of buildings, especially residential buildings, is a critical factor in the two earthquakes. All the affected areas of the Lushan earthquake and the Jiuzhaigou earthquake are situated in the old earthquake area of the Wenchuan earthquake of 2008. Since then, various seismic measures have been taken for new public and rural residential buildings. This has played a very important role in protecting people's lives and property safety [43]. According to the site conditions in the disaster area, the houses built by farmers were more damaged in the Lushan earthquake due to unreasonable structure or failure to take corresponding seismic measures meeting the requirements. In the Jiuzhaigou earthquake, the most densely populated area was the scenic spot. The houses rebuilt in the scenic area have good seismic performance and are only slightly damaged, effectively protecting people's lives and property. Therefore, enhancing the seismic performance of buildings helps to reduce their vulnerability to large earthquakes.

The seismic design standards of Chinese buildings are related to the local seismic fortification intensity. This indicator comprehensively considers the seismic environment, the importance of the construction project, the allowable risk level, the safety objectives to be achieved and the economic affordability. It can be said that buildings that meet the seismic fortification standards can better cope with the local earthquake disasters that may occur. However, in the above case analysis, it can be found that some buildings in the low-risk areas of earthquake disasters, as well as rural residential buildings, have not been designed and constructed in strict accordance with the seismic fortification standards. This has led to the fact that the actual seismic performance of the building cannot meet the requirements and has caused hidden dangers to people's lives and health and property safety. Therefore, the government should pay more attention to the implementation of standards in addition to the formulation of standards, such as strengthening the supervision of architectural design and construction manufacturing.

## 5. Conclusions

Earthquake disasters have brought great losses to human society. In this study, the DEA method is used to measure the vulnerability caused by earthquake disasters, and the variable of energy released by earthquakes is included in the input variables. Moreover, we use the statistical data of county-level units to calculate vulnerability, different from previous studies using provincial statistical data. This approach has the following advantages: On the one hand, using the DEA method to calculate vulnerability does not require setting the weights of various variables, which can avoid the impact of subjectivity on results. On the other hand, compared to the DEA models in the existing literature on seismic vulnerability, we introduce the energy released by earthquakes into the input variables. Then, we explore the influencing factors of vulnerability caused by earthquake disasters from the quantitative and qualitative perspectives, deepening the understanding of vulnerability. We selected 69 earthquake disasters in the Chinese mainland from 2013 to 2020 as samples for empirical research. The main findings are as follows:

(1) Four variables, including the added value of the secondary industry, per capita GDP, investment density in fixed assets and energy released by earthquakes, have a significant impact on pure technology vulnerability caused by earthquake disasters. If other conditions remain unchanged, increasing the added value of the secondary industry can reduce local vulnerability to earthquake disasters, while the growth of per capita GDP and investment density in fixed assets will increase vulnerability. With the rise in earthquake magnitude, the vulnerability caused by it will also become bigger.

(2) When suffering from an earthquake with M ≤ 5.0, some counties show a relatively high pure technical vulnerability. That is, there are deficiencies in organization, management and facilities. The areas with low earthquake risk show greater vulnerability when encountering earthquakes. Compared with regions with higher earthquake risk, they have lost their advantages of natural conditions and do not match the local socio-economic development level.

(3) Through the qualitative analysis of earthquake disaster cases, it can be inferred that the seismic function of buildings is related to vulnerability caused by earthquakes. Buildings that fully meet the seismic fortification standards play an important role in reducing earthquake disaster losses. However, buildings with problems in the seismic fortification design and construction will bring hidden dangers to people's lives and property safety.

Based on the above findings, some new enlightenment can be brought to the work of earthquake prevention and disaster reduction. First, all regions can appropriately expand the added value of the secondary industry according to the local situation. From the research results, increasing the per capita GDP and fixed asset investment density will increase vulnerability to earthquake disasters, which indicates that the adaptability of the work of earthquake disaster prevention and economic development should be further enhanced. Second, counties show higher levels of vulnerability when suffering from

earthquakes of M $\leq$ 5.0 than 5.0 < M $\leq$ 6.0. This reflects the insufficient attention paid to counties with a low risk of earthquake disasters. It is necessary to attach importance to and strengthen the work of earthquake prevention and disaster reduction in low-risk areas. Third, for provinces with frequent earthquakes, the seismic fortification of buildings should be further strengthened with high-quality standards. At the same time, attention should be paid to the implementation of standards in the design and construction stages of buildings in areas with low seismic risk, so as to avoid small earthquakes causing major disasters.

Earthquakes are inevitable. Reducing regional vulnerability to earthquake disasters can make the regional loss smaller. Only in this way can regions continuously enhance their sustainable development capabilities. In this study, the DEA method is used to accurately assess the vulnerability of counties to earthquake disasters, which is conducive to finding deficiencies and making targeted improvements.

This research extends the calculation methods of vulnerability caused by earthquake disasters, including the energy released by earthquakes as an important variable. Based on the analysis of the vulnerability results and earthquake disaster cases, the important role of building seismic function is further elaborated. In addition, the work of this research proves that regional vulnerability to natural disasters is related to their types, which can inspire the vulnerability research of various fields. However, due to the limitation of data collection, only eight potential influencing factors were tested. Further research is needed to determine the impact of regional natural condition factors on vulnerability caused by earthquakes.

**Author Contributions:** Methodology, Y.G.; software, Y.G.; formal analysis, Y.G., X.Y. and M.X.; investigation, Y.G; writing—original draft preparation, Y.G.; writing—review and editing, Y.G., X.Y., M.X. and Q.Z.; supervision, Q.Z.; project administration, Q.Z.; funding acquisition, Q.Z. All authors have read and agreed to the published version of the manuscript.

**Funding:** This research was funded by Qiuhong Zhao OF FUNDER, National Natural Science Foundation of China, under Project Nos. 72174019 and 72021001.

**Institutional Review Board Statement:** Not applicable.

**Informed Consent Statement:** Not applicable.

**Data Availability Statement:** The earthquake data are released by the official website of the China Earthquake Administration. The statistical data on the population and economy in counties are from the China County Statistical Yearbook from 2013 to 2020.

**Conflicts of Interest:** The authors declare no conflict of interest.

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
