# Peer review of "Assessment of Vulnerability Caused by Earthquake Disasters Based on DEA: A Case Study of County-Level Units in Chinese Mainland"

_sustainability, doi:10.3390/su15097545_

Round 1

Reviewer 1 Report

The paper needs to be revised. Please, check the comments in the attached PDF file. 

Author Response

Thank you very much for your useful comments. You raised the following important issues.

  1. The acronyms in abstract including CCR, BCC and GDP should be declared.

Response: Thanks for the comment. By following your suggestion, we have added the full names of GDP in the Abstract. CCR and BCC are two classic DEA models named with the author's initials. Generally, we think it can be used directly without the need to indicate full name.

  1. I suggest to frame the problem in a more specific way, considering that the method to investigate vulnerability at large scale are different.

There are empirical methods, mechanical methods and hybrid methods. Here some suggestions about the above methods, which authors could enclose in the revised version of the manuscript.

-empirical - 10.1016/j.ijdrr.2020.101807 (A prioritization RVS methodology for the seismic risk assessment of RC school buildings)

- mechanical - 10.1007/s10518-022-01516-7 (Analytical-mechanical based framework for seismic overall fragility analysis of existing RC buildings in town compartments)

- hybrid - 10.1007/s10518-022-01385-0 (Seismic vulnerability assessment of minor Italian urban centres: development of urban fragility curves)

Response: Thanks for the suggestion. Our research aims to quantify the vulnerability of the region to earthquake disaster and explore the influencing factors. It belongs to the evaluation research. We have carefully read the three articles you provided. We believe that our research about assessing the regional vulnerability to earthquake disaster is in a hybrid way. We have added the description of these three methods in Section 1, and thus introduced the methods and computational processes used in this research. The papers you suggested have been cited in this revised version. Thank you.

  1. Acronyms in 2.1 are not defined.

Response: Thanks for the comment. We carefully checked the abbreviations in Section 2.1. Both DEA and DMU were given their full names when they first appeared in Section 1.

  1. Take care to the format. It seems that it is an equation and it misses of the eq. number.

Response: Thanks for the suggestion. We have checked the formula in the entire text and added serial number of each equation in the manuscript.

  1. At this point, it is not clear how did authors employed DEA method for seismic vulnerability. In addition, about data, no sufficient information was provided. A map or an indication of the used database should be provided.

Response: Thanks for the suggestion. In order to enhance the explanation of how to apply DEA in earthquake vulnerability assessment, we have done the following work. Firstly, in Section 1, we added some literature on using the DEA method to calculate vulnerability, which can demonstrate the rationality of the method. Secondly, in Section 2, before introducing the specific model, we added the principle of calculating vulnerability through DEA models and input-output indicators.

The DEA method can calculate the efficiency of different decision-making units (which are counties in this study), i.e. the relative level of input-output ratio. When a county encounters earthquake disaster, it can be regarded as a disaster bearing system. It converts earthquake energy, population and property in the area into personnel and direct economic losses. When the input level is the same, the larger the output, the higher the system conversion efficiency, and the higher the regional vulnerability to earthquake disaster.

In terms of data, in this revised version, we have added the sources of data in Section 3. This study investigates the earthquake disasters that occurred in Chinese mainland from 2013 to 2020. The earthquake data is released by the official website of the China Earthquake Administration. And the statistical data on population and economy in counties are from the China County Statistical Yearbook from 2013 to 2020.

  1. Also the description of the application is extremely scarce. What are the parameters did authors consider on their dataset to employ the method?

Response: Thanks for the question. Regarding the application of the DEA method, based on your previous suggestion, we have explained it more clearly in this revised version. When we use DEA method to calculate the vulnerability of counties to earthquake disaster, the key is to determine the input variables and output variables. In this regard, we mainly rely on the concept of vulnerability and analysis of the disaster transformation process. At the same time, we referred to the variables used in other studies. In Section 2.2, we have supplemented the process of selecting variables.

  1. Considering that authors assumed empirical data, what about specific features of buildings in the area investigated? For sure, different classes of buildings present different vulnerability and then, they should be treated differently.

Response: Thanks for the question. As you mentioned, buildings with different materials, structures and construction times often show different vulnerabilities in earthquake disaster. We did consider architectural factors in our research. For one thing, we use case analysis to discuss the effect of seismic function of buildings on vulnerability caused by earthquakes in Section 4.2. For another, the input variables of the vulnerability assessment model include local economic level, which can reflect the construction situation to some extent.

  1. In the end, it is not clear what is the main advantage of this approach, especially if compared with other analogous existing ones. I suggest to better describe this aspect.

Response: Thanks for the suggestion. We elaborate the advantages of using DEA method to calculate the vulnerability of counties suffering from earthquake disaster from the following two aspects. On the one hand, compared to other methods, using the DEA method to calculate vulnerability does not require setting the weights of various variables, which can avoid the impact of subjectivity on the results. On the other hand, compared to the DEA models in existing literature on seismic vulnerability, we introduce the energy released by earthquakes into the input variables. Besides, we consider vulnerability at the county level. Therefore, the model can be more consistent with the definition of vulnerability, and the evaluation results can better reflect the role of seismicity. We have added the above explanation in Section 5.

Reviewer 2 Report

In this article the authors analyze 69 disasters produced by earthquakes in China, using the DEA method. The introduction contains the appropriate context to understand the subsequent analysis, however, I think it would be very useful to improve the methodology part by showing some explicit example of the calculation of the vulnerability measures used in this work.

In section 2.2 they allude to the magnitude as a measure of energy released by an earthquake, but they do not indicate what magnitude they are going to use for this study, is it Ml, Mc, Mw? Can you clarify what magnitude you use?

The condition that is in lines 120 and 121 is not clear.

I would appreciate it if you separate the tables from the text that comes after them.

About the statement in line 387 "small earthquakes and major disasters" what about the depth of the events? Did they consider it in this analysis?

Did you analyze what happens with the aftershocks? what about the vulnerability when there are aftershocks?

Minor bugs:

In lines 308 to 310, 345 to 347, 373 to 375 there is extra "the".

Author Response

Thank you very much for your useful comments. You raised the following important issues.

  1. In this article the authors analyze 69 disasters produced by earthquakes in China, using the DEA method. The introduction contains the appropriate context to understand the subsequent analysis. However, I think it would be very useful to improve the methodology part by showing some explicit examples of the calculation of the vulnerability measures used in this work.

Response: Thanks for your suggestion. To improve the methodology part by showing some explicit examples of the calculation, we have added the following contents. First, at the beginning of Section 2, we wrote a paragraph describing the process and principles of using DEA method to calculate vulnerability. In this way, readers can grasp how to use the method before reading the other sections. Second, we defined the corner markers of variables in Equation (1), which facilitates readers to correspond input variables and output variables one by one. When solving vulnerability values, only the data needs to be brought in. Third, we added the sources of data used to solve the model in Section 3. Also, Table 2 shows the vulnerability results calculated by substituting the data into Equation (1). We believe that these supplementary works can help readers have a better understanding of the solving process.

  1. In section 2.2 they allude to the magnitude as a measure of energy released by an earthquake, but they do not indicate what magnitude they are going to use for this study, is it Ml, Mc, Mw? Can you clarify what magnitude you use?

Response: Thanks for the comment. We use  in the study and we have made modifications in the revised version of the manuscript.

  1. The condition that is in lines 120 and 121 is not clear.

Response: Thanks for the comment. As we can see, lines 120 and 121 are Equation (1) describing the BCC model. The  in the objective function has been changed to , representing the vulnerability of . The constraint was originally to ensure that the weighted output of all DMUs does not exceed the weighted input, while maintaining the weight of each indicator unchanged. For the convenience of solving, it was linearly transformed, which is the expression in Equation (1).

  1. I would appreciate it if you separate the tables from the text that comes after them.

Response: Thanks for the suggestion. We have adjusted the layout of the table in the manuscript to make it more aesthetically pleasing and easy to read.

  1. About the statement in line 387 "small earthquakes and major disasters" what about the depth of the events? Did they consider it in this analysis?

Response: Thanks for the question. In the phrase "small earthquakes and big disasters", "small" and "large" are relative, compared with other earthquake disasters in magnitude and loss. Considering that the meaning of this phrase is not very clear, we have used a sentence instead in the revised manuscript, which is highlighted in Section 4.2 of the revised manuscript.

  1. Did you analyze what happens with the aftershocks? What about the vulnerability when there are aftershocks?

Response: Thanks for the question. For most recorded earthquakes, there are aftershocks in addition to the main shock. The magnitude is generally about the main earthquake, but the statistical earthquake disaster losses include the impact of aftershocks. In the DEA model we use, the output variables include the casualties and economic losses caused by earthquake disaster, which is influenced by aftershocks. Therefore, we considered aftershocks in vulnerability calculation.

Round 2

Reviewer 1 Report

The paper has been improved. I believe that it could be accepted for publication.